# Symptom Severity and Health-Related Quality of Life in Patients with Atrial Fibrillation: Findings from the Observational ARENA Study

**DOI:** 10.3390/jcm11041140

**Published:** 2022-02-21

**Authors:** Monika Sadlonova, Jochen Senges, Jonas Nagel, Christopher Celano, Caroline Klasen-Max, Martin Borggrefe, Ibrahim Akin, Dierk Thomas, Christopher Jan Schwarzbach, Thomas Kleeman, Steffen Schneider, Matthias Hochadel, Tim Süselbeck, Harald Schwacke, Angelika Alonso, Markus Haass, Karl-Heinz Ladwig, Christoph Herrmann-Lingen

**Affiliations:** 1Department of Psychosomatic Medicine and Psychotherapy, University of Göttingen Medical Center, 37075 Gottingen, Germany; jonas.nagel@med.uni-goettingen.de (J.N.); caroline.klasen@med.uni-goettingen.de (C.K.-M.); cherrma@gwdg.de (C.H.-L.); 2Department of Cardiovascular and Thoracic Surgery, University of Göttingen Medical Center, 37075 Gottingen, Germany; 3German Center for Cardiovascular Research (DZHK), Partner Site Gottingen, 37075 Gottingen, Germany; 4Department of Psychiatry, Massachusetts General Hospital, Boston, MA 02114, USA; ccelano@mgh.harvard.edu; 5Department of Psychiatry, Harvard Medical School, Boston, MA 02114, USA; 6Institute of Myocardial Infarction Research, Hospital of Ludwigshafen, 67063 Ludwigshafen, Germany; senges@stiftung-ihf.de (J.S.); schneider@stiftung-ihf.de (S.S.); hochadel@stiftung-ihf.de (M.H.); 7Department of Cardiology, Pneumology, Angiology, and Emergency Medicine, University of Mannheim Medical Center, 68167 Mannheim, Germany; martin.borggrefe@umm.de (M.B.); ibrahim.akin@umm.de (I.A.); 8German Center for Cardiovascular Research (DZHK), Partner Site Heidelberg/Mannheim, 69120 Heidelberg, Germany; dierk.thomas@med.uni-heidelberg.de; 9Department of Internal Medicine III—Cardiology, Angiology and Pneumology, Medical University, Hospital Heidelberg, 69120 Heidelberg, Germany; 10Department of Neurology, Hospital of Ludwigshafen, 67063 Ludwigshafen, Germany; schwarch@klilu.de; 11Hospital of Ludwigshafen, 67063 Ludwigshafen, Germany; kleemant@klilu.de; 12Clinic of Cardiology, 67071 Ludwigshafen, Germany; tim.sueselbeck@live.de; 13Diakonissen-Stiftungs-Hospital Speyer, 67346 Speyer, Germany; harald.schwacke@diakonissen.de; 14Department of Neurology, Mannheim Center for Translational Neuroscience, Medical Faculty Mannheim, University of Heidelberg, 68167 Mannheim, Germany; angelika.alonso@umm.de; 15Department of Cardiology, Theresien Hospital and St. Hedwig Clinic GmbH, 68165 Mannheim, Germany; m.haass@theresienkrankenhaus.de; 16Department of Psychosomatic Medicine and Psychotherapy, University Hospital Rechts der Isar, Technical University Munich, 81675 Munich, Germany; karl-heinz.ladwig@tum.de; 17German Centre for Cardiovascular Research (DZHK), Partner Site Munich Heart Alliance, 81675 Munich, Germany

**Keywords:** atrial fibrillation, EHRA class, health-related quality of life

## Abstract

Background: Atrial fibrillation (AF) is the most common sustained cardiac arrhythmia and is associated with impaired health-related quality of life (HRQoL), high symptom severity, and poor cardiovascular outcomes. Both clinical and psychological factors may contribute to symptom severity and HRQoL in AF. Methods: Using data from the observational Atrial Fibrillation Rhine-Neckar Region (ARENA) trial, we identified medical and psychosocial factors associated with AF-related symptom severity using European Heart Rhythm Association symptom classification and HRQoL using 5-level EuroQoL- 5D. Results: In 1218 AF patients (mean age 71.1 ± 10.5 years, 34.5% female), female sex (OR 3.7, *p* < 0.001), preexisting coronary artery disease (CAD) (OR 1.7, *p* = 0.020), a history of cardioversion (OR 1.4, *p* = 0.041), cardiac anxiety (OR 1.2; *p* < 0.001), stress from noise (OR 1.4, *p* = 0.005), work-related stress (OR 1.3, *p* = 0.026), and sleep disturbance (OR 1.2, *p* = 0.016) were associated with higher AF-related symptom severity. CAD (β = −0.23, *p* = 0.001), diabetes mellitus (β = −0.25, *p* < 0.001), generalized anxiety (β = −0.30, *p* < 0.001), cardiac anxiety (β = −0.16, *p* < 0.001), financial stress (β = −0.11, *p* < 0.001), and sleep disturbance (β = 0.11, *p* < 0.001) were associated with impaired HRQoL. Conclusions: Psychological characteristics, preexisting CAD, and diabetes may play an important role in the identification of individuals at highest risk for impaired HRQoL and high symptom severity in patients with AF.

## 1. Introduction

Atrial fibrillation (AF) is the most common sustained cardiac arrhythmia [1] and is associated with an increased long-term risk of stroke, heart failure, and all-cause mortality [2,3,4]. Although AF is asymptomatic in at least one-third of patients (‘silent AF’) [5], AF can lead to a broad range of symptoms, including palpitations, dizziness, fatigue/lack of energy, shortness of breath, or reduced exercise tolerance [6,7], which can lead to higher AF-related symptom severity [8] and impaired health-related quality of life (HRQoL) [8]. AF-related symptom severity is typically assessed in several ways, either reflecting objective duration and frequency of AF or functional severity of AF-related symptoms as assessed by the European Heart Rhythm Association (EHRA) symptom classification [9]. Rhythm control treatment decisions are influenced by symptom severity [10].

In patients with AF, both symptom severity and HRQoL may have important clinical implications. AF-related symptom severity is associated with increased healthcare utilization, including treatment procedures and hospitalization [11,12,13]. Moreover, AF patients show significantly impaired HRQoL in both physical and mental domains [8,14,15,16,17,18], and AF patients with impaired HRQoL show a higher risk of hospitalization [8]. Furthermore, prior studies reported a negative association between AF-related symptom severity (i.e., EHRA class) and HRQoL [8,9,13]. Unfortunately, these studies did not adjust for covariables such as age, sex, AF type, and comorbidities in their correlational analyses.

Identifying factors associated with symptom severity and impaired HRQoL in AF may be important for clinical AF management. Regarding symptom severity, higher EHRA class is associated with cardioversion, catheter ablation (CA), and heart failure incidence [18]. Furthermore, hyperlipidemia, hypertension, chronic obstructive pulmonary disease, thrombo-embolic complications, and perceived frequency and duration of AF are associated with higher self-rated AF-related symptom severity [19]. Women also show a substantially higher AF-related symptom severity [19] and lower health perception than men [17]. Psychological factors such as increasing severity of depression, anxiety, and somatization are associated with increasing AF-related symptom severity [20]. Moreover, poor sleep quality is highly prevalent in non-valvular AF patients and is related to AF-related symptom severity [21].

Medical and psychological factors are associated with impaired HRQoL as well. For example, higher body mass index (BMI), female gender, higher CHA_2_DS_2_-VASc score, and exposure to β-blockers and calcium channel blockers were associated with reduced HRQoL in a small study [22]. Additionally, anxiety, concerns regarding complications, uncertainty about the future, disease acceptance, AF symptom severity, higher perceived stress, and type D personality have been linked to impaired HRQoL in AF patients [15,22,23]. Further, a study showed that physicians rated their AF patients’ HRQoL higher than patients did, and this discordance was associated with depression, sleep disorder, physical inactivity, and the absence of significant concomitant cardiac disease [24]. Taken together, these findings suggest that the impact of AF on mental health may be underestimated in clinical settings, and greater attention to psychosocial factors may help to identify individuals at higher risk for greater symptom severity and reduced HRQoL [23].

Despite the studies mentioned above, the factors associated with AF-related symptom severity and HRQoL remain under-researched in comparison to the extensive number of studies analyzing the prevalence and clinical consequences of AF. Additional studies to examine psychological and medical correlates of HRQoL and both objective and subjective AF-related symptom severity in sufficiently large samples are needed to close this gap. In addition, HRQoL and AF-related symptom severity have rarely been studied as distinct but probably related subjective consequences of AF.

Accordingly, the aim of this study was to identify medical and psychological factors associated with AF-related symptom severity and impaired HRQoL in a large AF population. On the basis of previous research, we hypothesize, controlling for age, sex, and AF type, (a) an independent association between impaired HRQoL and higher AF-related symptom severity; (b) associations between higher AF symptom severity and impaired HRQoL with comorbidities and history of AF treatments (cardioversion, CA); and (c) associations between higher EHRA class and impaired HRQoL with sleep disturbance, higher perceived stress, and higher generalized and cardiac anxiety. The findings of this analysis may help to characterize those individuals at highest risk for high symptom severity and impaired HRQoL and to identify opportunities for screening or treatment that may reduce symptom burden and improve HRQoL in this population.

## 2. Materials and Methods

### 2.1. Study Design and Patient Population

The Atrial Fibrillation Rhine-Neckar Region (ARENA) Project is an observational study of the Foundation Institute for Myocardial Infarction Research (IHF) in cooperation with the Departments of Cardiology of the Hospitals in Ludwigshafen, Heidelberg, and Mannheim, as well as local resident cardiologists and the Heidelberg University Hospital (Department of Clinical Pharmacology and Pharmacological Epidemiology) in Germany. Its aim is to improve patient-centered care and prognosis, particularly in optimizing stroke prophylaxis in patients with AF [25]. Inclusion criteria for this study were residence in the polycentric Rhine-Neckar Metropolitan Region in Germany, confirmed diagnosis of atrial fibrillation, informed consent to participate in the ARENA-project, and age ≥ 18 years [25]. More than 5000 patients were screened, and 2777 patients were included in the ARENA study between August 2016 and December 2018. Of these, 1857 patients returned the baseline questionnaires. We analyzed baseline data of a subset of 1218 patients with complete data for the medical (except for the intake of metoprolol, new-onset AF, ejection fraction (EF), BMI, and HAS-BLED score) and psychosocial variables of interest.

Each subject’s demographic profile and cardiac health status (including symptom severity of AF) were assessed directly after enrollment at the recruiting sites (baseline assessment) and were documented in an electronic case report form (eCRF). Questionnaires related to medications, HRQoL, and psychosocial characteristics were then mailed to participants, who completed and returned them by mail. The cohort will be contacted for follow-up once a year for up to 10 years via postal surveys and standardized telephone interviews.

All research was performed in accordance with the ethical principles of the Declaration of Helsinki [26]. Written informed consent was obtained from all the participants. The study team provided sufficient time to the participants to make a decision regarding participation in the study. The study was approved by the Ethics Committee of the Rhineland-Palatine state Medical Association (#837.366.15) and by the Ethical Review Committee of the University of Heidelberg (#B-F-2016-051) and the Mannheim Medical Center (#2016-613N-MA) in Germany. The investigators registered the ARENA study on ClinicalTrials.gov (NCT02978248) on 30 November 2016 (https://clinicaltrials.gov/ct2/show/NCT02978248, accessed on 4 February 2022).

### 2.2. Assessments

Demographic and clinical variables (sex, age, coronary artery disease (CAD), diabetes mellitus (DM), stroke/transient ischemic attack (TIA), chronic kidney disease, CHA_2_DS_2_-VASc score, HAS-BLED score, BMI, EF, new-onset AF, AF type, history of cardioversion/CA, and intake of metoprolol) were assessed at baseline. AF-related symptom severity was assessed by the EHRA symptom classification [27], which was validated and improved by the EHRA in 2014 [9]. This classification system assesses whether AF-related symptoms are present and to what extent they interfere with daily activities (EHRA class 1 ‘no symptoms´, EHRA 2 ´mild symptoms´, EHRA 3 ´severe symptoms´, and EHRA 4 ´disabling symptoms´) [28]. According to the 2020 guidelines of the European Society of Cardiology (ESC), the assessment of EHRA symptoms is of clinical relevance for treatment decisions [10].

Psychosocial factors were assessed with a questionnaire designed specifically for this study. It included items assessing educational and occupational status, different sources of stress (work, home, financial, noise), sleep disturbance, generalized anxiety disorder (GAD) symptoms using the GAD-2 questionnaire [29], and two items from the German version of the Cardiac Anxiety Questionnaire (CAQ-2) [30]. The two items of GAD-2 and CAQ-2 were aggregated to sum scores ranging from 0 to 6 for GAD-2 and from 0 to 8 for CAQ-2. Stress related to home, home, work, and finances were measured with items developed for the INTERHEART study [31].

HRQoL was assessed with the German version of the EQ-5D-5L, a questionnaire for assessing health on five dimensions (mobility, self-care, usual activities, pain/discomfort, anxiety/depression) in five levels each [32]. The particular item values were converted into index-based values (utilities) using the German Value Set for the EQ-5D-5L [33], ranging from −0.661 to 1; higher scores indicate higher HRQoL.

### 2.3. Statistical Analyses

Sample characteristics are presented for the study population and split up by EHRA class. To avoid too many categories with too small numbers of observations, we combined EHRA classes 2a and 2b (i.e., mild to moderate symptoms) as well as EHRA classes 3 and 4 (severe to disabling symptoms) into a single category each. For dimensional variables, means and standard deviations are presented. As not all variables were distributed normally, groups were compared by Kruskal–Wallis tests for independent samples. For categorical variables, absolute numbers and percentages are displayed. Proportions are compared by chi-squared test.

Factors potentially associated with EHRA class were then analyzed using multinomial logistic regression. For analysis of HRQoL as measured by EQ-5D-5L, we used linear multiple regression. All three hypotheses were tested in a regression model controlling for age, sex, and AF type. For EHRA class, the parameters indicate whether the respective input variable affects the probability of having EHRA class 2 or EHRA class 3 or 4 rather than EHRA class 1 (asymptomatic, the reference category), independently of the other variables in the same model. Odds ratios are reported as indicators of effect size. For EQ-5D-5L, the parameter estimates indicate the change in raw values of EQ-5D-5L with each unit change of the respective predictor. As additional indicators of effect size, we report β-values for each parameter. These values were arrived at by z-transforming the continuous (but not the categorical) input variables and the outcome variable prior to model estimation. The reported β-values thus express how many standard deviations the EQ-5D-5L changes with one standard deviation change in the continuous input variables or with a change of categorical input variables from one level to the other, keeping all other variables in the model constant.

Finally, we performed a dropout analysis to compare the included subset of 1218 patients with confirmed AF and complete data for the variables of interest with dropout subset of 1557 with incomplete data, using chi-squared tests for categorical variables and Wilcoxon rank-sum tests for continuous variables.

All statistical analyses were performed with R, version 4.0.2.

## 3. Results

### 3.1. Sample Characteristics

We included *N* = 1218 patients with complete data for all analyzed variables (mean age 71.1 ± 10.5 years, 34.5% female). The most frequent cardiac comorbidity was CAD (75.4%) followed by DM (22.2%). A history of cardioversion was reported in 30.2% of the study population, and 18.4% of the AF patients had a history of CA. Descriptive characteristics for the full sample and for individuals in each EHRA class are displayed in Table 1. In bivariate comparisons, EHRA class was associated with sex, age, AF type, DM, CHA_2_DS_2_-VASc score, new-onset AF, history of cardioversion, history of ablation, generalized and cardiac anxiety, sleep disturbance, stress from work, home, and noise, as well as with HRQoL.

### 3.2. Association of AF-Related Symptom Severity and HRQoL

Regarding the first hypothesis, AF-related symptom severity was not associated with HRQoL when controlling for age, sex, and AF type. Table 2 presents parameters of the multiple regression model predicting EQ-5D-5L by EHRA class.

### 3.3. Medical and Psychological Correlates of AF-Related Symptom Severity

Testing the second hypothesis (Table 3), the probability of having EHRA class 3 or 4 rather than 1 increased with CAD (OR 1.7, *p* = 0.020). The probability of having EHRA class 2 (rather than 1) increased with history of cardioversion (OR 1.4, *p* = 0.041). Regarding the third hypothesis (Table 4), the probability of having EHRA class 2 rather than 1 increased with sleep disturbance (OR 1.2, *p* = 0.016), stress from noise (OR 1.4, *p* = 0.005), and CAQ-2 (OR 1.2, *p* < 0.001). The probability of having EHRA class 3 or 4 rather than 1 increased with higher levels of stress related to work (OR 1.3, *p* = 0.026) and CAQ-2 (OR 1.2, *p* < 0.001). In a combined multinomial logistic regression model of the significant medical and psychosocial variables from the Table 3 and Table 4, all the above-mentioned associations remained significant (Appendix A).

Furthermore, the probability of having higher EHRA class increased with female sex. EHRA class 3 or 4 (rather than 1) was associated with persistent (versus paroxysmal) AF (OR 1.6, *p* = 0.033).

### 3.4. Medical and Psychological Correlates of HRQoL

Testing the second hypothesis (Table 5), we found that CAD (β = −0.23, *p* = 0.001) and DM (β = −0.25, *p* < 0.001) were associated with impaired HRQoL. The test of the third hypothesis (Table 6) reported that GAD-2 (β = −0.30, *p* < 0.001), CAQ-2 (β = −0.16, *p* < 0.001), sleep disturbance (β = −0.11, *p* < 0.001), and financial stress (β = −0.11, *p* < 0.001) were associated with impaired HRQoL. In a combined linear multiple regression model of the significant medical and psychosocial variables from the Table 5 and Table 6, all the above-mentioned associations remained significant, except for the relationship between CAD and HRQoL (Appendix A).

Furthermore, impaired HRQoL was associated with permanent (vs. paroxysmal) AF (β = −0.26, *p* < 0.001). Although female sex was associated with HRQoL in the other regression models, this association disappeared in the multiple regression model, including psychosocial variables (Table 6).

### 3.5. Dropout Analyses

In comparison to the dropout subset of patients with incomplete data, included patients were younger than dropouts (71.1 ± 10.5 y. versus 73.7 ± 11.1 y., *p* < 0.001); less likely to be female (34.5% versus 39.8%, *p* = 0.004); slightly less likely to be single (21.3% versus 25.6%, *p* = 0.041); and less likely to have DM (22.2% versus 27.6%, *p* = 0.001), chronic kidney disease (16.6% versus 22.1%, *p* < 0.001), or stroke/TIA (11.6% versus 15.5%, *p* = 0.003). Included patients had a lower probability of being newly diagnosed with AF (10.2% vs. 16.6%, *p* < 0.001), slightly higher EF (*M* = 52.74, *SD* = 12.43 versus *M* = 51.08, *SD* = 13.46, *p* = 0.022), lower CHA_2_DS_2_-VASc (*M* = 3.27, *SD* = 1.65 versus *M* = 3.82, *SD* = 1.76, *p* < 0.001), and lower HAS-BLED scores (*M* = 2.00, *SD* = 1.06 versus *M* = 2.28, *SD* = 1.09, *p* < 0.001). Included patients had higher rates of cardioversion (30.2% versus 19.4%, *p* < 0.001) and CA (18.4% versus 10.7%, *p* < 0.001). All three patients *not* treated with metoprolol were included. EHRA class was higher in included patients than in dropouts (EHRA class 1: 39.7% versus 47.0%; EHRA class 2: 44.1% vs. 39.0%; EHRA class 3 or 4: 16.2% vs. 14.0%; *p* = 0.001). Finally, included patients reported higher levels of cardiac anxiety (*M* = 3.08, *SD* = 1.83 versus *M* = 2.80, *SD* = 1.82, *p* = 0.002) and work-related stress than dropouts (*M* = 0.49, *SD* = 0.83 versus *M* = 0.27, *SD* = 0.65, *p* < 0.001). There were no differences between the included and excluded populations on any of the remaining variables.

## 4. Discussion

We investigated the interrelation of AF-related symptom severity and HRQoL, as well as their associations with medical and psychosocial factors in a sample of 1218 AF patients. Our findings confirmed some—but not all—of our hypotheses. Consistent with the hypotheses, severe to disabling AF-related symptoms (EHRA class 3 or 4 rather than 1) were associated with CAD, perceived stress at work, and cardiac anxiety. Furthermore, mild- to-moderate symptoms (EHRA class 2 rather than 1) were associated with history of cardioversion, cardiac anxiety, sleep disturbance, and stress from noise. As expected, the probability of having higher EHRA class increases with female sex. HRQoL was associated with sex only when psychological variables were not controlled for. Furthermore, impaired HRQoL was associated with CAD, DM, sleep disturbance, financial stress, and generalized and cardiac anxiety. Contrary to our hypotheses, there was no independent association between AF-related symptom severity and HRQoL when adjusting for age, sex, and type of AF. In what follows, we draw on prior literature to outline hypotheses about causal mechanisms potentially underlying the observed relationships.

### 4.1. Relationship between AF-Related Symptom Severity and HRQoL

We found no evidence for a detrimental effect of increased AF-related symptom severity on HRQoL. Although one small study found that AF-related symptom severity was not an independent predictor of HRQoL [22], most studies in this area have found an inverse relationship between EHRA symptom class and HRQoL [8,9,13,34]. This includes two large trials: Schnabel et al. [13] demonstrated (*n* = 6196) that EHRA symptom classification and its components were moderately correlated with individual EQ-5D-5L items, and ORBIT-AF (*n* = 10,087) found an inverse association between EHRA class and disease-specific HRQoL assessed by the Atrial Fibrillation Effect on Quality-of-Life questionnaire (AFEQT) [8].

There are several possible explanations for this. First, the current study controlled for multiple potential confounding variables (e.g., age, sex, AF type), while many of the prior trials that found relationships between AF-related symptom severity and HRQoL did not. It is possible that factors not included in those analyses explained—at least in part—the relationship between symptom severity and HRQoL. Second, the current trial included a generic measure of HRQoL, and it may be that this measure did not fully capture HRQoL in patients with AF. Consistent with this idea, Wynn et al. [9] found that while EHRA class was negatively associated with both AF-specific and generic HRQoL, AF-specific HRQoL (measured with the AFEQT) was much more strongly associated with EHRA class than generic HRQoL. Similarly, Essebag et al. [34] suggested that assessments of AF-related QoL may be more specific and sensitive to changes in AF-related symptom severity compared to generic HRQoL scales. However, generic instruments such as the EQ-5D are better suited for cost-effectiveness analyses. It remains unclear as to why patients in our study with EHRA classes 3–4 tended to report even better HRQoL in bivariate analyses than those in lower EHRA classes. Research studies that include both generic and AF-specific measures of HRQoL and that adjust for relevant confounders may be helpful to clarify the relationship between AF-related symptom severity and HRQoL.

### 4.2. Factors Associated with Increased AF-Related Symptom Severity

Our findings concerning associations of medical and psychosocial factors with AF-related symptom severity appear consistent with existing literature. Next to medical factors such as history of cardioversion and CAD, prior work found that female sex [17,35,36,37,38] and lower age [37] are related to symptom severity in AF patients. In the RACE II and the FRACTAL studies, women reported more severe AF-related symptoms [39,40]. Several explanations have been described for this observation. First, women with AF have more cardiac comorbidities such as hypertension, heart failure with preserved ejection fraction, and valvular heart disease [36]. Second, longitudinal analyses have found that women with AF have greater impairments in mental health and tendencies for somatization than men, although women continued to experience worse physical function and greater symptom severity even after controlling for somatization [41]. There may be sex differences in expressing and coping with AF symptoms, which may have important implications for treatment of AF.

Several psychological conditions, such as anxiety, depression, or perceived stress, have also been linked to AF-related symptom severity. Charitakis et al. [42] reported that anxiety, left atrial dilatation, and low-grade inflammation were significant predictors of arrhythmia-related symptoms in patients with AF. Another study [43] found that there was an association between a higher AF-related symptom severity and increased anxiety or depression. Interestingly, in both adjusted and unadjusted follow-up analyses, antiarrhythmic drug therapy or CA reduced AF symptom severity, but neither the perception of AF frequency nor the severity of anxiety or depression improved significantly with AF treatment [43]. In another study [44] investigating patients with persistent AF planned for cardioversion, higher levels of emotional distress (anxiety, depression, and perceived stress) were significantly associated with the number and frequency of reported AF symptoms. The authors suggest implementing screening and treatment of emotional distress as a patient-centered approach into cardiological care to reduce attentional bias toward bodily sensations and to influence the success rate of cardioversion.

In our study, disturbed sleep was associated with EHRA class 2 (rather than 1) but not with EHRA class 3 or 4 (rather than 1). Obstructive sleep apnea is an established risk factor for AF, promoting arrhythmogenesis and impairing treatment efficacy [45]. There are only a few studies reporting relationships of other sleep-related problems with AF incidence, such as short sleep duration or frequent nocturnal awaking [46,47,48]. Concerning AF-related symptom severity in particular, Szymanski et al. [21] found that poor sleep quality is highly prevalent in AF patients, and that its prevalence increases with higher EHRA class. Sleep quality may play a role in the pathogenesis and prediction of AF, potentially representing a novel target for prevention or treatment. Alternatively, sleep disturbance may be a somatic symptom of depression or anxiety disorder. In our analyses, we could not adjust for depression, and our anxiety scale (GAD-2) did not assess sleep-related anxiety symptoms.

Finally, there is some evidence that specific psychosocial stressors such as occupational stress [49] or exposure to traffic noise [50] or noise annoyance [51] are risk factors for increased AF-related symptom perception; however, to our knowledge, their relationships with AF symptom severity have not yet been investigated. Further studies should confirm the associations between the AF symptom severity and work- or noise-related stress [52]. Whether psychosocial interventions could impact the symptom severity of AF though a reduction of work- and noise-related stress requires further research.

### 4.3. Factors Associated with Impaired HRQoL

In our study, preexisting CAD and DM were associated with impaired HRQoL. Both cardiac symptoms and increased heart-related attention in CAD patients might help to explain the relationships between CAD and HRQoL. Furthermore, the presence of DM was associated with impaired HRQoL. This finding is consistent with a study among 192 patients with AF in which DM was a negative predictor of improvement in HRQoL between baseline and 1 year follow-up [53]. Metabolic changes and coping with the chronic disease and its behavioral and health consequences long term might be possible explanations for the association between DM and impaired HRQoL in AF patients. Surprisingly, we found no associations between HRQoL and AF interventions such as cardioversion or CA. However, AF interventions were reported retrospectively, and their benefits may have vanished over time.

The key finding of this study is that both higher AF-related symptom severity and impaired HRQoL—although poorly interrelated—are associated with several psychological variables. Interventions that target generalized and cardiac anxiety, stress, and sleep disturbance may be important for improvement of HRQoL in AF patients. For example, in the SMURF study, symptoms of anxiety and depression significantly predicted poor HRQoL in patients with AF [42]. One mechanism underlying the predictive value of anxiety and depression for poor HRQoL might be the overestimation of duration and frequency of AF episodes [42]. Another explanation for impaired HRQoL due to anxiety or depression might be the cognitive interpretation (e.g., catastrophizing) of bodily AF sensations and dysfunctional coping with AF symptoms. Furthermore, a study on patients with paroxysmal AF showed that trait anxiety, psychological stress events, and anxiety symptoms were strong determinants of poor HRQoL [54]. Similarly, Ong et al. [55] showed that anxiety sensitivity was associated with poorer HRQoL, greater symptom severity, and increased distress in AF patients, and the tested models explained 19–40% of the variance in HRQoL and distress [55]. The authors recommended focusing more strongly on the psychosocial aspects in AF management, as those might be critical determinants of patient´s HRQoL [54,55].

From a clinical perspective, it may be useful to consider screening for psychosocial factors (e.g., stress; sleep quality; anxiety; and, specifically, cardiac anxiety) in patients with a diagnosis of AF, especially in cases of high AF-related symptom severity and impaired HRQoL. In cases of a positive psychosocial screen, the primary care providers or cardiologists could provide additional information (e.g., educational materials) about stress prevention, anxiety reduction, or sleep hygiene. Collaboration with psychosomatic/psychiatric specialists might be a next possible treatment option of persistent stress, anxiety, or sleep disorders in patients with AF. Such a holistic approach in AF management has the potential to reduce AF-related symptom severity and increase HRQoL.

Finally, an integrated approach including avoidance of stroke, better symptom management, and cardiovascular and comorbidity risk reduction (ABC pathway) could be a holistic way for clinical decision-making steps in AF management [56]. Such a novel approach might reduce AF-related symptom severity and increase HRQoL in AF patients.

### 4.4. Strengths and Limitations

This study has several strengths, the most important one being the large sample size of patients with AF. Another important strength is the concurrent analysis of both medical and psychosocial factors, as well as their associations with symptom severity and HRQoL in patients with AF. Furthermore, our sample consisted of a heterogeneous group of patients with paroxysmal, persistent, or permanent AF.

Notable limitations include the study’s cross-sectional design (which prevents the identification of causal relationships), absence of a comparator group of participants without AF, and measurement of HRQoL using a generic scale (EQ-5D-5L) rather than an AF-specific one. In addition, the effect of depression as a potential predictor of symptom severity and HRQoL would have been of interest, but there was no measure of depression in this study. Furthermore, we did not consider data regarding oral anticoagulation from the ARENA study, and we did not adjust our statistical analyses for all important clinical AF factors (e.g., comorbidities or use of oral anticoagulation), which might have led to bias in our results. Finally, the analyzed subset showed several differences from the excluded population including age, sex, marital status, chronic kidney disease, stroke/TIA, newly diagnosed AF, EF, CHA_2_DS_2_-VASc score, HAS-BLED score, EHRA class, cardioversion, CA, DM, cardiac anxiety, and stress related to work. Therefore, the results should be generalized with caution. However, included and excluded patients were similar with respect to the remaining variables.

## 5. Conclusions

In conclusion, the novelty of this study is a multidimensional approach using medical data, history of AF treatments, and psychosocial characteristics and their associations with AF-related symptom severity and HRQoL. Our results indicate that it might be important to consider psychosocial factors such as generalized and cardiac anxiety, sleep disturbance, work-related stress, and stress from noise as possibly affecting AF-related symptom severity and HRQoL. Such a holistic approach in AF management has the potential to reduce AF-related symptom severity and increase HRQoL, and it might inform clinical decisions for invasive treatments.

However, the relationships between psychosocial predictors and symptom severity of AF are still unclear and require longitudinal data analyses. Finally, the impact of psychosocial variables, age, sex, and factors of medical history on HRQoL should be better understood for optimizing the clinical treatment algorithms in AF patients and to improve their HRQoL.

## Figures and Tables

**Table 1 jcm-11-01140-t001:** Sample characteristics of the study population, both overall and split up by EHRA class.

EHRA Class											
	Total (*n* = 1218 *)		1 (*n* = 484)		2 (*n* = 537)		3 or 4 (*n* = 197)				
Categorical Variables	*n*	*%*	*n*	%	*n*	%	*n*	%	χ²	*df*	*p*
Sex (female)	420	34.5	123	25.4	194	36.1	103	52.3	45.897	2	<0.001
Single	255	21.3	98	20.6	114	21.6	43	22.3	0.274	2	0.872
University entrance qualification	646	54.0	247	52.0	299	56.6	100	51.5	2.703	2	0.259
CAD	918	75.4	371	76.7	394	73.4	153	77.7	2.144	2	0.342
DM	271	22.2	122	25.2	100	18.6	49	24.9	7.315	2	0.026
AF type									15.760	4	0.003
Paroxysmal	742	60.9	288	59.5	336	62.6	118	59.9			
Persistent	291	23.9	100	20.7	136	25.3	55	27.9			
Permanent	185	15.2	96	19.8	65	12.1	24	12.2			
History of cardioversion	368	30.2	123	25.4	180	33.5	65	33.0	8.796	2	0.012
History of CA	224	18.4	66	13.6	112	20.9	46	23.4	12.694	2	0.002
Metoprolol *	1074	99.7	451	100.0	472	99.4	151	100.0	3.813	2	0.149
Chronic kidney disease	202	16.6	80	16.5	85	15.8	37	18.8	0.910	2	0.634
Stroke/TIA	141	11.6	57	11.8	63	11.7	21	10.7	0.193	2	0.908
New-onset AF *	124	10.2	41	8.5	53	9.9	30	15.3	7.084	2	0.029
**Continuous Variables**	** *M* **	** *SD* **	** *M* **	** *SD* **	** *M* **	** *SD* **	** *M* **	** *SD* **	** *H* **	** *df* **	** *p* **
Age	71.1	10.5	73.5	9.4	69.9	10.7	68.8	11.3	41.225	2	<0.001
EF *	52.74	12.43	51.58	12.60	53.35	12.34	53.73	12.18	4.708	2	0.095
BMI *	28.16	5.17	27.79	4.85	28.43	5.31	28.32	5.54	2.263	2	0.323
CHA_2_DS_2_-VASc	3.27	1.65	3.39	1.58	3.14	1.68	3.29	1.74	6.919	2	0.031
HAS-BLED *	2.00	1.06	2.10	1.02	1.96	1.05	1.90	1.16	5.644	2	0.059
GAD-2	1.46	1.36	1.21	1.29	1.63	1.37	1.59	1.40	30.173	2	<0.001
CAQ-2	3.08	1.83	2.66	1.89	3.34	1.72	3.43	1.77	47.324	2	<0.001
Sleep disturbance	1.14	0.88	0.99	0.85	1.24	0.88	1.25	0.90	28.058	2	<0.001
Stress at work	0.49	0.83	0.38	0.76	0.52	0.83	0.67	0.96	17.645	2	<0.001
Stress at home	1.01	0.58	0.93	0.55	1.07	0.60	1.05	0.60	14.225	2	0.001
Financial stress	0.32	0.55	0.31	0.53	0.33	0.56	0.32	0.56	0.286	2	0.867
Stress from noise	0.48	0.63	0.38	0.57	0.56	0.67	0.51	0.64	21.448	2	<0.001
EQ-5D-5L	0.80	0.25	0.80	0.27	0.80	0.23	0.82	0.23	5.973	2	0.050

*Note.* AF = atrial fibrillation; BMI = body mass index; CA = catheter ablation; CAD = coronary artery disease; CAQ-2 = Cardiac Anxiety Questionnaire 2-item screener; DM = diabetes mellitus; EF = ejection fraction; EQ-5D-5L = EuroQoL-5D; GAD-2 = Generalized Anxiety Disorder 2-item screener; TIA = transient ischemic attack. Differences by EHRA class in categorical variables were tested with χ² tests. Differences in continuous variables were tested with Kruskal–Wallis rank sum tests. * The total number of analyzed participants was 1218 with complete data for medical and psychosocial variables, except of intake of metoprolol (*n* = 1077), new-onset AF (*n* = 1212), EF (*n* = 705), BMI (*n* = 1198), and HAS-BLED score (*n* = 1209).

**Table 2 jcm-11-01140-t002:** Parameters of multiple regression model predicting EQ-5D-5L by EHRA class.

Hypothesis	Predictor	Estimate	95% CI	β	*t*-Value	*p*
	(Intercept)	1.025	0.927; 1.123	0.183	20.401	<0.001
Confounders	Age	−0.003	−0.005; −0.001	−0.107	−3.714	<0.001
	Sex (female)	−0.079	−0.108; −0.050	−0.321	−5.333	<0.001
	Persistent AF	0.012	−0.021; 0.045	0.050	0.743	0.457
	Permanent AF	−0.102	−0.141; −0.063	−0.416	−5.123	<0.001
H1	EHRA class 2	−0.017	−0.046; 0.012	−0.068	−1.097	0.273
	EHRA class 3/4	0.013	−0.028; 0.054	0.054	0.633	0.527

*Note.* AF = atrial fibrillation; EHRA = European Heart Rhythm Association; H1 = Hypothesis 1.

**Table 3 jcm-11-01140-t003:** Parameters of multinomial logistic regression model predicting EHRA class by medical status.

		EHRA Class 2a or 2b					EHRA Class 3 or 4				
	Predictor	Coef.	95% CI	*z*	*p*	OR	Coef.	95% CI	*z*	*p*	OR
	(Intercept)	2.240	1.288; 3.191	4.614	<0.001	9.392	1.696	0.479; 2.913	2.732	0.006	5.452
Confounders	Age	−0.035	−0.049; −0.021	−5.047	<0.001	0.966	−0.052	−0.070; −0.035	−5.779	<0.001	0.949
	Sex (female)	0.568	0.288; 0.847	3.980	<0.001	1.764	1.307	0.945; 1.668	7.080	<0.001	3.694
	AF persistent	0.192	−0.121; 0.505	1.203	0.229	1.212	0.387	−0.028; 0.801	1.826	0.068	1.472
	AF permanent	−0.360	−0.723; 0.013	−1.894	0.058	0.698	−0.256	−0.787; 0.274	−0.948	0.343	0.774
H2	CAD	0.169	−0.144; 0.483	1.059	0.290	1.184	0.520	0.081; 0.959	2.323	0.020	1.682
	DM	−0.258	−0.571; 0.056	−1.611	0.107	0.773	0.115	−0.295; 0.525	0.548	0.584	1.121
	Stroke/TIA	0.133	−0.264; 0.530	0.659	0.510	1.143	0.027	−0.531; 0.586	0.095	0.924	1.028
	Cardioversion	0.303	0.013; 0.593	2.045	0.041	1.353	0.237	−0.154; 0.627	1.188	0.235	1.267
	CA	0.246	−0.105; 0.598	1.372	0.170	1.279	0.386	−0.065; 0.837	1.679	0.093	1.471

*Note.* AF = atrial fibrillation; CA = catheter ablation; CAD = coronary artery disease; DM = diabetes mellitus; EHRA = European Heart Rhythm Association, H2 = Hypothesis 2; TIA = transient ischemic attack.

**Table 4 jcm-11-01140-t004:** Parameters of multinomial logistic regression model predicting EHRA class by psychosocial variables.

		EHRA Class 2a or 2b					EHRA Class 3 or 4				
	Predictor	Coef.	95% CI	*z*	*p*	OR	Coef.	95% CI	*z*	*p*	OR
	(Intercept)	1.784	0.637; 2.931	3.051	0.002	5.954	0.895	−0.585; 2.375	1.186	0.236	2.447
Confounders	Age	−0.038	−0.054; −0.022	−4.965	<0.001	0.963	−0.045	−0.065; −0.025	−4.528	<0.001	0.956
	Sex (female)	0.376	0.082; 0.670	2.508	0.012	1.456	1.181	0.805; 1.557	6.145	<0.001	3.258
	AF persistent	0.252	−0.066; 0.570	1.554	0.120	1.286	0.453	0.036; 0.870	2.126	0.033	1.572
	AF permanent	−0.418	−0.792; −0.044	−2.183	0.029	0.658	−0.211	−0.740; 0.318	−0.779	0.436	0.810
H3	Sleep disturbance	0.205	0.038; 0.372	2.416	0.016	1.228	0.208	−0.013; 0.429	1.831	0.067	1.231
	Stress—work	0.031	−0.153; 0.215	0.332	0.740	1.032	0.260	0.031; 0.489	2.221	0.026	1.296
	Stress—home	0.086	−0.169; 0.341	0.658	0.510	1.089	−0.029	−0.362; 0.304	−0.169	0.866	0.972
	Stress—financial	−0.478	−0.754; −0.202	−3.386	0.001	0.620	−0.473	−0.843; −0.103	−2.509	0.012	0.623
	Stress—noise	0.326	0.099; 0.553	2.810	0.005	1.385	0.176	−0.130; 0.482	1.132	0.258	1.192
	GAD-2	0.098	−0.025; 0.221	1.560	0.119	1.104	0.024	−0.139; 0.187	0.283	0.777	1.024
	CAQ-2	0.159	0.077; 0.241	3.800	<0.001	1.172	0.196	0.086; 0.306	3.527	<0.001	1.216

*Note.* AF = atrial fibrillation; CAQ-2 = Cardiac Anxiety Questionnaire 2-item screener; EHRA = European Heart Rhythm Association; GAD-2 = Generalized Anxiety Disorder 2-item screener; H3 = Hypothesis 3.

**Table 5 jcm-11-01140-t005:** Parameters of multiple regression model predicting EQ-5D-5L by medical status.

Hypothesis	Predictor	Estimate	95% CI	β	*t*-Value	*p*
	(Intercept)	1.011	0.917; 1.105	0.389	20.984	<0.001
Confounders	Age	−0.002	−0.003; 0.000	−0.068	−2.313	0.021
	Sex (female)	−0.083	−0.111; −0.054	−0.336	−5.732	<0.001
	Persistent AF	0.016	−0.016; 0.048	0.065	0.971	0.332
	Permanent AF	−0.081	−0.122; −0.042	−0.330	−4.027	<0.001
H2	CAD	−0.056	−0.089; −0.023	−0.227	−3.332	0.001
	DM	−0.062	−0.095; −0.030	−0.253	−3.726	<0.001
	Stroke/TIA	−0.039	−0.081; 0.003	−0.159	−1.828	0.068
	Cardioversion	0.003	−0.027; 0.033	0.013	0.208	0.836
	CA	0.004	−0.032; 0.040	0.017	0.227	0.820

*Note.* AF = atrial fibrillation; CA = catheter ablation; CAD = coronary artery disease; DM = diabetes mellitus; H2 = Hypothesis 2; TIA = transient ischemic attack.

**Table 6 jcm-11-01140-t006:** Parameters of multiple regression model predicting EQ-5D-5L utilities by psychosocial variables.

Hypothesis	Predictor	Estimate	95% CI	β	*t*-Value	*p*
Confounders	(Intercept)	1.303	1.205; 1.401	0.054	26.190	<0.001
	Age	−0.004	−0.006; −0.002	−0.163	−5.990	<0.001
	Sex (female)	−0.012	−0.037; 0.013	−0.047	−0.908	0.364
	Persistent AF	0.001	−0.026; 0.028	0.006	0.098	0.922
	Permanent AF	−0.064	−0.097; −0.031	−0.260	−3.753	<0.001
H3	Sleep disturbance	−0.030	−0.044; −0.016	−0.106	−4.017	<0.001
	Stress—work	0.012	−0.004; 0.028	0.039	1.475	0.140
	Stress—home	−0.014	−0.036; 0.008	−0.034	−1.260	0.208
	Stress—financial	−0.049	−0.073; −0.025	−0.110	−4.044	<0.001
	Stress—noise	−0.019	−0.039; 0.001	−0.050	−1.936	0.053
	GAD-2	−0.055	−0.065; −0.045	−0.304	−10.192	<0.001
	CAQ-2	−0.021	−0.029; −0.013	−0.158	−5.916	<0.001

*Note.* AF = atrial fibrillation; CAQ-2 = Cardiac Anxiety Questionnaire 2-item screener; GAD-2 =Generalized Anxiety Disorder 2-item screener; H3 = Hypothesis 3.

## Data Availability

The ARENA dataset generated and/or analyzed for the present article is available upon request to the corresponding author.

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
