# Peer review of "Symptom Severity and Health-Related Quality of Life in Patients with Atrial Fibrillation: Findings from the Observational ARENA Study"

_jcm, 2022, doi:10.3390/jcm11041140_

Round 1

Reviewer 1 Report

This interesting study investigated whether the medical and psychological factors are associated with AF-related symptom severity and impaired HRQoL in the AF population. Overall, the manuscript is well-written.  I have a few recommendations that I think may improve the manuscript.

Comments:

Methodology:

 2.1. Study design and patient population

Also, write the country name in the first paragraph.

Define ARENA……Project Atrial Fibrillation Rhine-Neckar Region.

As per the records of clinicaltrials.gov, actual enrolment was 2,776 participants between August 2016 and December 2018.

“Inclusion criteria for this study were residence in the area” Not clear enough. was it Rhein-Neckar Region in Germany? Please clarify.

 It is not clear how many participants received and returned the questionnaires related to medications, HRQoL, and psychosocial characteristics. I suggest specifying this information, as well.

I recommend revising the following statement “All research was performed in accordance with relevant guidelines and regulations.” Are there any specific national guidelines or regulations? It seems vague when we only write ‘relevant’.

With reference to the participants’ recruitment, did the research team provide sufficient time to the participants to taking decisions regarding their participation?

“The investigators registered on ClinicalTrials.gov (NCT02978248) on……………..” What did they register? Not clear enough.

2.2. Assessments

EHRA was earlier defined in the introduction. Kindly avoid duplication.

The authors stated that the psychosocial factors were assessed with a questionnaire designed specifically for this study. Has this tool been validated?

Results

Also, specify the inferential statistical tests in the footnote of Table 1.

Discussion

Using the terms higher cardiac anxiety and increased cardiac anxiety may confuse the readers.

4.1. Relationship between AF-related symptom severity and HRQoL

I suggest avoiding the word ‘surprisingly’.

Charitakis at al…………….Its et al………Please correct it.

“that increased anxiety or depression were associated with increased AF symptom severity after adjusting for potential confounders”…………This statement was taken from the following study without reasonable paraphrasing https://onlinelibrary.wiley.com/doi/full/10.1111/pace.12292 . Please rephrase it.

Author Response

Comments:

Methodology:

2.1. Study design and patient population

Comment: Also, write the country name in the first paragraph.

Comment: Define ARENA……Project Atrial Fibrillation Rhine-Neckar Region.

Answer: Thank you for this remark. In response to these comments, we now define the ARENA project and added the country name on p. 3 as follows and : “The Atrial Fibrillation Rhine-Neckar Region (ARENA) Project is an observational study of the Foundation Institute for Myocardial Infarction Research (IHF) in cooperation with the Departments of Cardiology of the Hospitals in Ludwigshafen, Heidelberg, and Mannheim, as well as local resident cardiologists and the Heidelberg University Hospital (Department of Clinical Pharmacology and Pharmacological Epidemiology) in Germany.”

Comment: As per the records of clinicaltrials.gov, actual enrolment was 2,776 participants between August 2016 and December 2018.

Answer: Thank you for your important remark. We will correct the number of enrolled participants on clinicaltrials.gov. The correct enrollment was 2,777 participants between August 2016 and December 2018.

Comment: “Inclusion criteria for this study were residence in the area” Not clear enough. was it Rhein-Neckar Region in Germany? Please clarify.

Answer: Thank you for your valuable comment. We clarified the inclusion criteria on p. 3 as follows: “Inclusion criteria for this study were residence in the polycentric Rhine-Neckar Metropolitan Region in Germany, confirmed diagnosis of atrial fibrillation, informed consent to participate in the ARENA-project, and age ≥ 18 years [25].”

Comment: It is not clear how many participants received and returned the questionnaires related to medications, HRQoL, and psychosocial characteristics. I suggest specifying this information, as well.

Answer: Thank you for this important comment. 1,857 participants returned the baseline questionnaire related to medications, psychosocial characteristics, and medical data. We analyzed baseline data of a subset of 1,218 patients with complete data for the medical and psychosocial variables of interest. We added the following aspects into the manuscript on p. 3: “More than 5,000 patients were screened, and 2,777 patients were included in the ARENA study between August 2016 and December 2018. Of these, 1,857 patients returned the baseline questionnaires.”

Comment: I recommend revising the following statement “All research was performed in accordance with relevant guidelines and regulations.” Are there any specific national guidelines or regulations? It seems vague when we only write ‘relevant’.

Answer: Thank you for your helpful remark. We added the following clarification on p. 3: “All research was performed in accordance with the ethical principles of the Declaration of Helsinki [26].”

Comment: With reference to the participants’ recruitment, did the research team provide sufficient time to the participants to taking decisions regarding their participation?

Answer: Thank you for your question. Yes, the study team provided sufficient time to the study participants to make a decision regarding the study participation. We added the following sentence on p. 3 to our manuscript: “The study team provided sufficient time to the participants to make a decision regarding participation in the study.”

Comment: “The investigators registered on ClinicalTrials.gov (NCT02978248) on……………..” What did they register? Not clear enough.

Answer: Thank you for the remark. We clarified the sentence on p. 3 as follows: “The investigators registered the ARENA study on ClinicalTrials.gov (NCT02978248) on November, 30th 2016 https://clinicaltrials.gov/ct2/show/NCT02978248.”

2.2. Assessments

Comment: EHRA was earlier defined in the introduction. Kindly avoid duplication.

Answer: Thank you for the remark. We deleted the EHRA definition on p. 3 as suggested: “AF-related symptom severity was assessed by EHRA symptom classification [27], which was validated and improved by the EHRA in 2014 [9].”

Comment: The authors stated that the psychosocial factors were assessed with a questionnaire designed specifically for this study. Has this tool been validated?

Answer: Thank you for the important question. Stress was assessed using the published items from the INTERHEART study [31]. In that case-control study, 11,119 patients with a first myocardial infarction and 13,648 age- and sex-matched controls were analyzed. The psychosocial stressors were associated with increased risk of acute myocardial infarction in this study population, which can be considered an indicator of validity. Furthermore, generalized anxiety disorder (GAD) was assessed using the validated GAD-2 screener [29], and heart focused anxiety was assessed using two key items from the validated German version of the Cardiac Anxiety Questionnaire [30], covering the aspects of heart-related fear (1 item) and avoidance (1 item). In a previous study (Edelmann et al.  2017), we found that the fear item used in ARENA correlated at r=0.76 (n=289) with the CAQ subscale “fear,” while the avoidance item correlated at r=0.77 (n=291) with the CAQ subscale “avoidance.” Finally, single purpose-designed items were used to assess sociodemographic factors and sleep disturbance.

Results

Comment: Also, specify the inferential statistical tests in the footnote of Table 1.

Answer: Thank you for your valuable comment. We added the following aspects into the footnote of the Table 1 on p. 5: “Differences by EHRA class in categorical variables were tested with χ²-tests. Differences in continuous variables were tested with Kruskal-Wallis rank sum tests.”

Discussion

Comment: Using the terms higher cardiac anxiety and increased cardiac anxiety may confuse the readers.

Answer: Thank you for this valuable remark. We corrected the aspects on p. 9 as follows: “Consistent with the hypotheses, severe to disabling AF-related symptoms (EHRA class 3 or 4 rather than 1) were associated with CAD, perceived stress at work, and cardiac anxiety. Furthermore, mild to moderate symptoms (EHRA class 2 rather than 1) were associated with history of cardioversion, cardiac anxiety, sleep disturbance, and stress from noise.”

4.1. Relationship between AF-related symptom severity and HRQoL

Comment: I suggest avoiding the word ‘surprisingly’.

Answer: Thank you for the comment. We deleted the word ´surprisingly´ on p. 9 in the Discussion section.

Comment: Charitakis at al…………….Its et al………Please correct it.

Answer: Thank you for the remark. We corrected the mistake on p. 10.

Comment: “that increased anxiety or depression were associated with increased AF symptom severity after adjusting for potential confounders”…………This statement was taken from the following study without reasonable paraphrasing https://onlinelibrary.wiley.com/doi/full/10.1111/pace.12292 . Please rephrase it.

Answer: Thank you for the comment. We rephrased the sentence on p. 10 as follows: “Another previous study [43] found that there was an association between a higher AF-related symptom severity and increased anxiety or depression. Interestingly, in both adjusted and unadjusted follow-up analyses, antiarrhythmic drug therapy or CA reduced AF symptom severity, but neither the perception of AF frequency nor the severity of anxiety or depression improved significantly with AF treatment [43].”

References:

  1. World Medical Association. World Medical Association Declaration of Helsinki: Ethical Principles for Medical Research Involving Human Subjects. JAMA. 2013, 310, 2191–2194, doi:10.1001/jama.2013.281053

  2. Kroenke, K.; Spitzer, R.L.; Williams, J.B.W.; Monahan, P.O.; Löwe, B. Anxiety disorders in primary care: prevalence, impairment, comorbidity, and detection. Ann. Intern. Med. 2007, 146, 317–325; doi:10.7326/0003- 4819-146-5-200703060-00004.

  1. Eifert, G.H.; Thompson, R.N.; Zvolensky, M.J.; Edwards, K.; Frazer, N.L.; Haddad, J.W.; Davig, J. The Cardiac Anxiety Questionnaire: development and preliminary validity. Behaviour Research and Therapy 2000, 38, 1039– 1053, doi:10.1016/S0005-7967(99)00132-1.

  1. Rosengren, A.; Hawken, S.; Ôunpuu, S.; Sliwa, K.; Zubaid, M.; Almahmeed, W.A.; Blackett, K.N.; Sitthi-amorn, C.; Sato, H.; Yusuf, S. Association of psychosocial risk factors with risk of acute myocardial infarction in 11 119 cases and 13 648 controls from 52 countries (the INTERHEART study): case-control study. The Lancet 2004, 364, 953–962, doi:10.1016/S0140-6736(04)17019-0.

Edelmann, F.; Bobenko, A.; Gelbrich, G.; Hasenfuss, G.; Herrmann-Lingen, C.; Duvinage, A., Schwarz, S., Mende, M.; Prettin, C.; Trippel, T.; Lindhorst, R.; Morris, D.; Pieske-Kraigher, E.; Nolte, K.; Düngen, H.D.; Wachter, R.; Halle, M.; Pieske, B. Exercise training in Diastolic Heart Failure (Ex-DHF): rationale and design of a multicentre, prospective, randomized, controlled, parallel group trial. Eur J Heart Fail. 2017, 19, 1067-1074, doi: 10.1002/ejhf.862.

Reviewer 2 Report

Major comments:

1) Sample characteristics of the study population are very limited; no data on AF management, new-onset AF, or comorbidities.

2) The logistic regression analysis is adjusted for age, sex, and AF type. Why did you choose these variables? The results might be biased as the other clinically important factors were not included. 

3) No data on oral anticoagulation; it would add to the study to compare the OAC-treated vs not-anticoagulated patients, as well as vitamin K vs non-vitamin K antagonists. 

4) What is novel in the study? And what's your clinical perspective? You stated in the conclusion:

"it might be important to consider psychosocial factors such as generalized and cardiac anxiety, sleep disturbance, work-related stress, and stress
from noise possibly affecting AF-related symptom severity, which might in turn inform clinical decisions for invasive treatments";

- but you do not present any follow-up data. It would be interesting to see the follow-up. 

Minor comments:

1) The novel management of AF should be based on the ABC pathway (Lip GYH. The ABC pathway: an integrated approach to improve AF management. Nat Rev Cardiol. 2017 Nov;14(11):627-628. doi: 10.1038/nrcardio.2017.153. Epub 2017 Sep 29. PMID: 28960189.). It would be worth to mention in the discussion - this novel approach may increase the patients' quality of life.  

Author Response

Replies:

Comment: If one of the referees has suggested that your manuscript should undergo extensive English revisions, please address this issue during revision. We propose that you use one of the editing services.

Answer: Thank you for your remark. We provided an extensive English revision. All the changes are marked in red.

Major comments:

Comment 1: Sample characteristics of the study population are very limited; no data on AF management, new-onset AF, or comorbidities.

Answer: We appreciate your remark. In response to the comment, in addition to the already included comorbidities (coronary artery disease, diabetes mellitus), we have added information about several relevant comorbidities, including chronic kidney disease, and stroke/transient ischemic attack. Additionally, we have added CHA2DS2-VASc and HAS-BLED scores to better characterize participants’ medical status related to AF. Furthermore, we added information about body mass index, ejection fraction, and new-onset AF. Finally, in addition to the already included characteristics of AF management (i.e., history of cardioversion and catheter ablation), we now report use of metoprolol at the baseline assessment. These additional variables are also considered in the dropout analyses.

Comment 2: The logistic regression analysis is adjusted for age, sex, and AF type. Why did you choose these variables? The results might be biased as the other clinically important factors were not included. 

Answer: Thank you for your important question. There are several reasons for the inclusion of age, sex, and AF type as covariates in the regression analyses. We included age because the prevalence of atrial fibrillation increases with advanced age (Staerk et al., 2017), and higher age and frailty are associated with increased symptom severity in AF patients (Wilkinson et al., 2019). We included sex as a covariate, as AF is more prevalent among men (Staerk et al., 2017), and women with AF experience substantially higher AF-related symptom severity [19] and reduced HRQoL [22] compared to men. We included AF type, as perceived frequency and duration of AF have been linked to higher self-rated AF-related symptom severity [19], and longer AF episodes are associated with impaired QoL (Jansson et al. 2021).

Additionally, we included stroke/TIA in our regression models predicting EHRA class (Table 3), and HRQoL (Table 5) by medical status. Furthermore, we now present a combined multinomial logistic regression model of the significant medical and psychosocial variables from the Table 3 and Table 4 with EHRA (Supplement 1). All associations with EHRA remained significant, indicating that the effects obtained in the individual analyses are mutually independent. Similarly, we additionally report a combined linear multiple regression model of the significant medical and psychosocial variables with HRQoL from the Table 5 and Table 6 (Supplement 2). Again, all associations (except the relationship between CAD and HRQoL) remained significant. We added the following aspects in our manuscript on  p. 6: “In a combined multinomial logistic regression model of the significant medical and psychosocial variables from the Table 3 and Table 4, all the above mentioned associations remained significant (Supplement 1).”, and on p. 8: “In a combined linear multiple regression model of the significant medical and psychosocial variables from the Table 5 and Table 6, all the above mentioned associations remained significant, except for the relationship between CAD and HRQoL (Supplement 2).”

Comment 3: No data on oral anticoagulation; it would add to the study to compare the OAC-treated vs not-anticoagulated patients, as well as vitamin K vs non-vitamin K antagonists. 

Answer: Thank you for your valuable comment. While we agree that an additional analysis of oral anticoagulation might be of importance/clinical relevance, we did not plan to report that information in the current manuscript. We are in the process of creating of a separate manuscript with a primary focus on AF management from the ARENA study and feel that information about oral anticoagulation and vitamin K antagonists would align with the aim of that manuscript more closely. We would highly appreciate your understanding.

Comment 4: What is novel in the study? And what's your clinical perspective? You stated in the conclusion:

"it might be important to consider psychosocial factors such as generalized and cardiac anxiety, sleep disturbance, work-related stress, and stress
from noise possibly affecting AF-related symptom severity, which might in turn inform clinical decisions for invasive treatments";

- but you do not present any follow-up data. It would be interesting to see the follow-up. 

Answer: Thank you for important questions. We believe that one key novelty of this study is a multi-dimensional approach using medical data, history of AF treatments (such as cardioversion, catheter ablation or medication), and several psychosocial characteristics and their associations with AF-related symptom severity and HRQoL. This provides an opportunity to consider AF-related symptom severity not only as a function of the somatic characteristics but as a bio-psycho-social consideration in patients with AF.               Our findings have several important clinical implications. The finding that psychosocial factors (e.g., perceived stress, anxiety) may be associated with higher AF-related symptom severity and impaired HRQoL suggests that screening for these factors (e.g., stress, sleep quality, anxiety and specifically cardiac anxiety) may be important in patients with a diagnosis of AF, especially in case of a high AF-related symptom severity and impaired HRQoL. Second, if a patient with AF has several characteristics associated with high AF symptom burden, the clinical team may intervene to address them. This may include the provision of additional information (e.g., educational materials) about stress prevention, strategies for anxiety reduction, or sleep hygiene. For individuals with persistent stress, an anxiety disorder, or a sleep disorder, referral to a psychosomatic/psychiatric specialist might be appropriate. We will provide follow-up information from this observational study regarding HRQoL once it becomes available.                We added the following aspects into the conclusions on p. 12: “In conclusion, the novelty of this study is a multidimensional approach using medical data, history of AF treatments as well as psychosocial characteristics and their associations with AF-related symptom severity and HRQoL. Our results indicate that it might be important to consider psychosocial factors such as generalized and cardiac anxiety, sleep disturbance, work related stress, and stress from noise possibly affecting AF-related symptom severity, which might in turn inform clinical decisions for invasive treatments. From the clinical perspective, it may be useful to consider screening for these psychosocial factors (e.g., stress, sleep quality, anxiety and specifically cardiac anxiety) in patients with a diagnosis of AF, especially in case of a high AF-related symptom severity and impaired HRQoL. In case of a positive psychosocial screen, the primary care providers or cardiologists could provide additional information (e.g., educational materials) about stress prevention, anxiety reduction, or sleep hygiene. Collaboration with psychosomatic/psychiatric specialists might be a next possible treatment option of persistent stress, anxiety or sleep disorders in patients with AF. Such a holistic approach in AF management has the potential to reduce AF-related symptom severity and increase HRQoL.”

Minor comments:

Comment 1: The novel management of AF should be based on the ABC pathway (Lip GYH. The ABC pathway: an integrated approach to improve AF management. Nat Rev Cardiol. 2017 Nov;14(11):627-628. doi: 10.1038/nrcardio.2017.153. Epub 2017 Sep 29. PMID: 28960189.). It would be worth to mention in the discussion - this novel approach may increase the patients' quality of life.  

Answer: Thank you for your important suggestion. We mentioned the novel AF management based on an ABC pathway in the Discussion section on p. 11 as follows: “Finally, an integrated approach including avoidance of stroke, better symptom management, and cardiovascular and comorbidity risk reduction (ABC pathway) could be a holistic way for clinical decision-making steps in AF management [56]. Such a novel approach might reduce AF-related symptom severity and increase HRQoL in AF patients.”

References:

  1. Eisenhart Rothe, A. von; Hutt, F.; Baumert, J.; Breithardt, G.; Goette, A.; Kirchhof, P.; Ladwig, K.-H. Depressed mood amplifies heart-related symptoms in persistent and paroxysmal atrial fibrillation patients: a longitudinal analysis--data from the German Competence Network on Atrial Fibrillation. Europace 2015, 17, 1354–1362, doi:10.1093/europace/euv018.

  1. Walters, T.E.; Wick, K.; Tan, G.; Mearns, M.; Joseph, S.A.; Morton, J.B.; Sanders, P.; Bryant, C.; Kistler, P.M.; Kalman, J.M. Symptom severity and quality of life in patients with atrial fibrillation: Psychological function outweighs clinical predictors. Int. J. Cardiol. 2019, 279, 84–89, doi:10.1016/j.ijcard.2018.10.101.

  1. Lip, G.Y.H. The ABC pathway: an integrated approach to improve AF management. Nat. Rev. Cardiol. 2017, 14, 627-628, doi: 10.1038/nrcardio.2017.153.

Jansson, V., Bergfeldt, L., Schwieler, J., Kennebäck, G., Rubulis, A., Jensen, S.M., et al. Atrial fibrillation burden, episode duration and frequency in relation to quality of life in patients with implantable cardiac monitor. Int J Cardiol Heart Vasc. 2021, 11, 34, 100791, doi: 10.1016/j.ijcha.2021.100791.

Staerk, L.; Sherer, J.A.; Ko, D.; Benjamin, E.J.; Helm, R.H. Atrial Fibrillation: Epidemiology, Pathophysiology, and Clinical Outcomes. Circ Res. 2017, 28, 1501-1517, doi: 10.1161/CIRCRESAHA.117.309732.

Wilkinson, C.; Todd, O.; Clegg, A.; Gale, C.P.; Hall, M. Management of atrial fibrillation for older people with frailty: a systematic review and meta-analysis. Age Ageing. 2019, 1, 48, 196-203, doi: 10.1093/ageing/afy180.

Round 2

Reviewer 2 Report

Thank you for your hard work and revisions. The manuscript has been greatly improved. 

Minor comments:

  • I understand you do not want to share full data regarding anticoagulation in the study population. Please mark it as a study limitation that oral anticoagulation was not considered - as it is the only known factor improving survival in AF patients.
  • Please add to the study limitation that results might be biased as they are not adjusted for all-important clinical factors (including comorbidities, OAC use etc) 
  • Please re-redact the conclusion section, and put the "clinical perspective" to the discussion section.

Author Response

Comment: Thank you for your hard work and revisions. The manuscript has been greatly improved. 

Answer: Thank you very much for your remark and the improvement of our manuscript through your valuable comments.

Minor comments:

Comments:
a) I understand you do not want to share full data regarding anticoagulation in the study population. Please mark it as a study limitation that oral anticoagulation was not considered - as it is the only known factor improving survival in AF patients.

  1. b) Please add to the study limitation that results might be biased as they are not adjusted for all-important clinical factors (including comorbidities, OAC use etc) 

Answers: Thank you for these important comments. We added the following study limitations on p. 12: “Furthermore, we did not consider data regarding oral anticoagulation from the ARENA study, and we did not adjust our statistical analyses for all important clinical AF factors (e.g., comorbidities or use of oral anticoagulation) which might have led to bias in our results.”

Comment: Please re-redact the conclusion section, and put the "clinical perspective" to the discussion section.

Answer: Thank you for your remark. We now re-redacted the conclusion section on p. 13 as follows: “In conclusion, the novelty of this study is a multidimensional approach using medical data, history of AF treatments as well as psychosocial characteristics and their associations with AF-related symptom severity and HRQoL. Our results indicate that it might be important to consider psychosocial factors such as generalized and cardiac anxiety, sleep disturbance, work related stress, and stress from noise possibly affecting AF-related symptom severity and HRQoL. Such a holistic approach in AF management has the potential to reduce AF-related symptom severity and increase HRQoL, and it might inform clinical decisions for invasive treatments. However, the relationships between psychosocial predictors and symptom severity of AF are still unclear and require longitudinal data analyses. Finally, the impact of psychosocial variables, age, sex and factors of medical history on HRQoL should be better understood for optimizing the clinical treatment algorithms in AF patients and to improve their HRQoL.”

Furthermore, as suggested we moved the “clinical perspective” to the discussion section on p.12.
